# A Comparison of Tuning Methods for Predictive Functional Control

**John Anthony Rossiter \*** and **Muhammad Saleheen Aftab**

Department of Automatic Control and System Engineering, University of Sheffield, Mappin Street, Sheffield S1 3JD, UK; Msaftab1@sheffield.ac.uk
\* Correspondence: j.a.rossiter@sheffield.ac.uk

**Abstract:** Predictive functional control (PFC) is a fast and effective controller that is widely used in preference to PID for single-input single-output processes. Nevertheless, the core advantages of simplicity and low cost come alongside weaknesses in tuning efficacy. This paper summarises and consolidates the work of the past decade, which has focused on proposing more effective tuning approaches while retaining the core attributes of simplicity and low cost. The paper finishes with conclusions on the more effective approaches and links to context.

**Keywords:** predictive control; challenging dynamics; tuning; stability properties

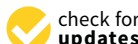



## 1. Introduction

Predictive functional control (PFC) is often used in preference to a PID approach on mainly single-input single-output (SISO) applications [1–4]. The reason industrial users might prefer PFC to PID is threefold: (i) being model-based, in theory at least, it can exploit the model information better and thus handle challenging dynamics; (ii) being based on prediction, constraint handling can be embedded in a systematic fashion and (iii) the coding complexity is similar to PID, which is elementary [5,6]; thus, maintenance and implementation are straightforward. A further important point that follows from the above three is that PFC is cheap (similar costs to PID) and is far cheaper than more conventional model predictive control (MPC) schemes, such as DMC and GPC [7].

Despite its widespread adoption in parts of the industry, a traditional PFC algorithm still has a number of weaknesses, with the most important one being that the tuning parameters are effective for only a limited range of dynamical systems and thus critically:

- Links between the tuning parameters and behaviour are not as intuitive as they need to be for many cases [8].
- For systems with challenging dynamics, a traditional PFC algorithm may fail to achieve satisfactory behaviour.

MPC theoreticians may also worry about the lack of a priori stability guarantees [9,10], but given that PFC is a competitor with PID, this issue is not important in practice, and it is common to use a posteriori stability checks.

The originators of PFC [6] proposed a number of ad hoc modifications to the basic algorithm to improve properties and tuning for systems with non-simple dynamics (e.g., open-loop integrators, instability, under-damping, non-minimum phase characteristics). The most popular proposal was to deploy a type of cascade structure, where an inner loop uses proportional only control to improve the dynamics for an outer loop to control with PFC [6,11,12]. This restriction makes it difficult to have a systematic selection of gain, and moreover, the consequent tuning proposals embed dynamics and assumptions that are somewhat contradictory of the original PFC concepts and, for many cases, the back-calculation used for constraint handling is equally ad hoc (suboptimal). In summary, it is difficult to see a systematic design behind these modifications, and the tuning still lacked intuition.

Consequently, in recent years, many authors have proposed a number of more systematic modifications to improve tuning for specific cases, e.g., [9,10,13–17]. Nevertheless, there is overlap in the application and benefits of these approaches alongside an undesirable multiplicity of options. Thus it is timely to write an overview paper that extracts the most useful aspects of these papers and provides some unified and narrower systematic guidance to the user.

Section 2 introduces conventional PFC for completeness before Section 3 summarises concisely the alternative approaches from recent years. The main contribution of this paper is in Section 4, which summarises the strengths and weaknesses of the various proposals before tabulating a recommended generic approach and offering some numerical examples for completeness.

## 2. Overview of Traditional PFC

This section presents, in brief, the main concepts, notation, and formulation of PFC [5,6,8,18]. As the focus is on the SISO case, transfer function-based models are used; thus, for example, assuming discrete time, the model will take the form:

$$a(z)y_k = b(z)u_k + \frac{\zeta_k}{\Delta(z)}; \quad \Delta(z) = 1 - z^{-1} \tag{1}$$

where $b(z) = b_1 z^{-1} + \dots$, $a(z) = 1 + a_1 z^{-1} + \dots$, $y_k, u_k$ are the outputs and inputs, respectively, at sample $k$, and $\zeta_k$ is an unknown zero mean random variable used to capture uncertainty (parametric and system disturbances).

**Remark 1.** *As this paper is focused on tuning and concepts, we will avoid explicit inclusion of disturbance handling in the algebra hereafter to simplify the presentation. The most common way of doing this in PFC (and also common in mainstream MPC) is simulating an independent model in parallel and using the difference between the independent model and the process outputs as a disturbance estimate $d_k$ to correct for any steady-state bias in the predictions. Here, as seen in Figure 1, $d_k = y_p - y_m$, with $y_p$ being the process output measurement and $y_m$ the independent model output.*

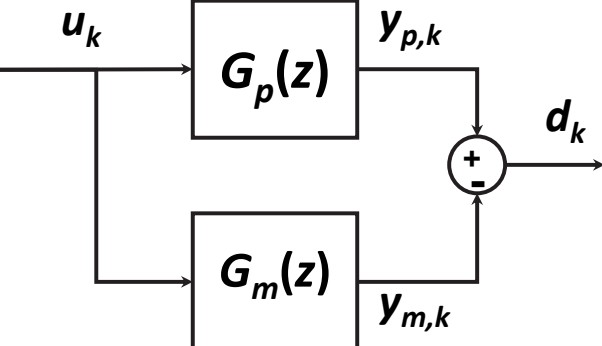

**Figure 1.** Independent model structure and disturbance estimate.

### 2.1. Prediction

Finding ouput predictions for model (1) is standard in the literature [7,19], so here we assume the reader is comfortable with the following result based on input increments $\Delta u_{k+i} = u_{k+i} - u_{k+i-1}$:

$$\underset{\rightarrow}{y}_{k+1|k} = H\underset{\rightarrow}{\Delta u}_k + P\underset{\leftarrow}{\Delta u}_k + Q\underset{\leftarrow}{y}_k \tag{2}$$

$$\underset{\rightarrow}{\Delta u}_k = \begin{bmatrix} \Delta u_k \\ \Delta u_{k+1} \\ \vdots \\ \Delta u_{k+n-1} \end{bmatrix}; \underset{\leftarrow}{\Delta u}_k = \begin{bmatrix} \Delta u_{k-1} \\ \Delta u_{k-2} \\ \vdots \\ \Delta u_{k-m} \end{bmatrix}; \underset{\leftarrow}{y}_k = \begin{bmatrix} y_k \\ y_{k-1} \\ \vdots \\ y_{k-m} \end{bmatrix}; \underset{\rightarrow}{y}_{k+1|k} = \begin{bmatrix} y_{k+1} \\ y_{k+2} \\ \vdots \\ y_{k+n} \end{bmatrix}$$

The parameters *H, P, Q* depend on the model parameters and are easy to determine. It is convenient to define a shorthand to extract individual rows of (2), so:

$$\underset{\rightarrow}{y}_{k+n|k} = H_n \underset{\rightarrow}{\Delta u}_k + P_n \underset{\leftarrow}{\Delta u}_k + Q_n \underset{\leftarrow}{y}_k \tag{3}$$

where $\mathbf{e}_n$ is the nth standard basic vector and $H_n = \mathbf{e}_n^T H, P_n = \mathbf{e}_n^T P, Q_n = \mathbf{e}_n^T Q$.

### 2.2. Conventional PFC Control Law

PFC design is intuitive in that one uses a first-order dynamic as a model for an ideal closed-loop response [18]; the associated time constant or pole being the most important design parameter. In practice, one aims to achieve this behaviour by ensuring an explicit matching between the predictions and 'ideal behaviour' at a single point, denoted the coincidence point. Hence, a PFC control law is defined by the equality:

$$y_{k+n|k} = (1 - \lambda^n)r + \lambda^n y_k \tag{4}$$

where *r* is the set point, $y_{k+n_y|k}$ is the *n*-step ahead system prediction, and $\lambda$ is the desired closed-loop pole (in effect $\lambda = e^{\frac{-3T}{T_s}}$, where *T* is the sampling period and $T_s$ is the desired settling time so practitioners could equivalently choose $T_s$ directly).

A core tenant of PFC is computational simplicity [6,18], so within the predictions we assume only a single degree of freedom (d.o.f.). Hence, selecting $\Delta u_{k+i} = 0$, $i > 0$ and combining this with prediction (3) and control law (4) gives:

$$\underbrace{H_n \mathbf{e}_1}_{h_n} \Delta u_k + P_n \underset{\leftarrow}{\Delta u}_k + Q_n \underset{\leftarrow}{y}_k = (1 - \lambda^n)r + \lambda^n y_k \tag{5}$$

Thus, after rearrangement, the control law (4) is implemented by solving:

$$\Delta u_k = \frac{1}{h_n} \left[ (1 - \lambda^n)r + \lambda^n y_k - Q_n \underset{\leftarrow}{y}_k - P_n \underset{\leftarrow}{\Delta u}_k \right] \tag{6}$$

### 2.3. Efficacy of Conventional PFC

Control law (6) works well when the open loop system has behaviour that is close to a monotonic step response, such as 1st order systems; for these cases, the tuning parameter $\lambda$ is then reasonably effective. However, for systems with more complex dynamics or significant lag in the initial response, the tuning parameter is much less effective [8,14,15]. The developments summarised in this paper are focused towards the latter cases.

### 2.4. Constraint Handling

Constraint handling is not a main discussion point in this paper, so it is included briefly here for completeness. The back calculation [20] favoured in traditional PFC papers is somewhat simplistic and suboptimal, so where constraint handling is required, the authors would recommend users to adopt approaches similar to those in mainstream MPC [14,19,21,22] whereby inequalities are developed to explicitly check every prediction point against the corresponding constraint.

$$\underline{\Delta u} \le \Delta u_k \le \overline{\Delta u}; \quad \underline{u} \le u_k \le \overline{u}; \quad \underline{y} \le y_k \le \overline{y}, \quad \forall k \tag{7}$$

For a suitably long horizon, constraints (7) can be captured by the inequalities:

$$C\Delta u_k \le \mathbf{f}_k \tag{8}$$

$$
C = \begin{bmatrix} 1 \\ -1 \\ 1 \\ -1 \\ H\mathbf{e_1} \\ -H\mathbf{e_1} \end{bmatrix} ; \quad
\mathbf{f}_k = \begin{bmatrix} \overline{u} - u_{k-1} \\ -\underline{u} + u_{k-1} \\ \overline{\Delta u} \\ -\underline{\Delta u} \\ L\overline{y} - P\underleftarrow{\Delta u}_k - Q\underleftarrow{y}_k \\ -L\underline{y} + P\underleftarrow{\Delta u}_k + Q\underleftarrow{y}_k \end{bmatrix} ; \quad
L = \begin{bmatrix} 1 \\ 1 \\ \vdots \\ 1 \end{bmatrix}
$$

where $\mathbf{f}_k$ depends on past data in $\underleftarrow{\Delta u}_k, \underleftarrow{y}_k$ and on the limits.

The input/output predictions will satisfy constraints if inequalities (8) are satisfied, and thus the PFC algorithm should consider these explicitly. Given there is a single d.o.f., a simple *for loop* within the code can find the choice of $\Delta u_k$ closest to (6) to ensure this very efficiently and, where appropriate, to ensure recursive feasibility properties (assuming convergent predictions).

**Remark 2.** *Small modifications to the algebra above are needed for the algorithms in the following sections, but given this is straightforward and does not require new concepts, the details are omitted.*

### 3. Summary of Recent Proposed Enhancements

This section outlines a number of proposals that have appeared in the literature to improve the tuning of efficacy of PFC. The fundamental problem [8,15] is that for many systems there is a poor correspondence between the chosen $\lambda$ and the resulting closed-loop pole, thus undermining a core selling point of PFC, that is, ease of tuning. Notably, for a system with open-loop unstable poles, significant underdamping, or integrating dynamics, PFC is quite challenging to tune effectively [8], and moreover, the resulting divergent or oscillating predictions may give rise to infeasibility and/or robustness issues.

#### 3.1. Fundamental Weaknesses and Core Conceptual Proposal

A fundamental conceptual weakness of a simplistic PFC approach is that one is basing decisions on an open-loop prediction, which may have undesirable, possibly divergent, dynamics. In addition to creating significant issues with reliable constraint handling, using open-loop predictions can easily lead to ill-posed problems with consequent loss of performance [19]. This is well recognised even in mainstream MPC [23], although to some extent, the issue can be partially side-stepped by having multiple d.o.f. so that the optimised predictions have better dynamics during transients.

Hence, and perhaps ironically, while MPC methods use optimisation of predictions to find a nicely shaped prediction, the optimisation itself is likely to fail unless the parameterisation of the predictions within the optimisation already have appropriate shaping and, surprisingly, the default shapings in many standard MPC laws are poorly chosen for some scenarios. To be more specific, this is a big issue and far more evident where there are low numbers of d.o.f., such as with PFC.

Hence, the underlying philosophy in the proposed modifications to PFC rely on a different parameterisation of the d.o.f. to that shown in (3), that is, the d.o.f. in the predicted future input sequence $\underrightarrow{\Delta u}_k$ are not just the first increment $\Delta u_k$. The parameterisation chosen must have two key attributes:

1. It can be reduced to a single d.o.f. so that computation and coding is trivial, in line with a conventional PFC approach.
2. The associated predictions should be better aligned to the desired behaviour of (4) than open-loop predictions.

Hence, the main philosophy deployed in recent work is to pre-stabilise/pre-shape the output predictions so that the effect of unwanted open-loop poles on the predictions are alleviated [24–26].

### 3.2. Input Shaping PFC

This section summarises the concept of algebraic or explicit input shaping of the open-loop predictions. It relies on explicit pole cancellation within the predictions [22] and some neat algebra, which requires simultaneous equations of a similar complexity to computation of the predictions.

Assume that the system model has some undesirable modes denoted by $a^+(z)$, so:

$$\Delta y(z) = \frac{b(z)}{a(z)} \Delta u(z); \quad a(z) = a^-(z) a^+(z) \tag{9}$$

The challenge is to determine $\Delta \underset{\rightarrow k}{u}$ such that the implied $u(z)$ (including past behaviour), cancels the modes $a^+(z)$ from the future predictions. It can be shown [22] that such a parameterisation takes the form:

$$\Delta \underset{\rightarrow k}{u} = P_1 \mathbf{p}; \quad \mathbf{p} = A_1 \underset{\leftarrow k}{y} + A_2 \Delta \underset{\leftarrow k}{u} \tag{10}$$

for suitable $P_1, A_1, A_2$. One can add future d.o.f. to the predictions with an additional term as follows:

$$\Delta \underset{\rightarrow k}{u} = P_1 \mathbf{p} + C_{a^+} \phi \tag{11}$$

where the parameter $\phi$ denotes the degrees of freedom (d.o.f.) and $C_{a^+}$ is a Toeplitz matrix of the parameters in $a^+(z)$. The corresponding output predictions, from which all the modes linked to $a^+(z)$ are now absent, can be deduced using similar algebra to be:

$$\underset{\rightarrow k+1}{y} = P_2 \mathbf{p} + H_s \phi \tag{12}$$

for suitable matrices $P_2, H_s$.

**Remark 3.** *For PFC, one would choose $\phi$ to be a simple scalar, that is, with one d.o.f. More complex MPC algorithms may choose this to be a vector or indeed other forms such as $\phi(z)/\alpha(z)$ where $\alpha(z)$ contains some desirable closed-loop dynamics.*

To finish up, it is important to give some reflections on input shaping Algorithm 1 and whether this is an approach worth further investigation.

- On the positive side, the shaping is effective at removing undesirable modes from the predictions, which can be considered essential for reliable constraint handling and recursive feasibility for systems with unstable and/or oscillatory modes.
- On the downside, this approach does not help with the weak links between the tuning parameter $\lambda$ and the closed-loop behaviour.
- The explicit cancellation used in the predictions can lead to sensitive results [24–26] and potentially aggressive input trajectories. Indeed, this conceptual approach, while interesting, has not been pursued in the mainstream MPC literature. One can mitigate against the aggressive input sightly by incorporating the pole parameter $\alpha(z)$, but, as yet, no systematic guidance exists for this process and, as discussed later, one could argue better alternatives exist.
- The algebra required to produce the parameterisations in (11), (12) are not simple in general to code, albeit the code would be very quick to execute and could be written in about 50 lines of code (e.g., in C, Python). This mitigates against the core selling points of simplicity.

---

**Algorithm 1** The PFC input shaping law is derived by:

---

1. Substitute predictions (12) into (4) to solve for $\phi$.
2. Use (11) to determine the current system input $\Delta u_k$.

### 3.3. Pole Placement PFC

The pole placement technique [10] exploits properties of PFC associated with 1st order systems. It can be shown that for a first order system, the parameter $\lambda$ is a precise tuning parameter, that is, the nominal closed-loop will have a pole at $\lambda$ as requested. As before, this section avoids giving the detailed derivations as those are available in the original sources, and here, we focus on core concepts.

Using partial fraction expansions, one can represent a higher order process as a sum of first order processes, for example:

$$\frac{b(z)}{a(z)} = \frac{B_1}{z - A_1} + \frac{B_2}{z - A_2} + \cdots \tag{13}$$

The next trick is to find separate suitable PFC inputs for each of these parallel systems, which would drive the pole, for each system, to $\lambda$. A summation of these inputs applied to the actual system should also result in a system closed-loop pole being a $\lambda$, from simple superposition arguments. The formal algebra and derivations in the original sources shows that this principle works very well, and one can indeed achieve a closed-loop pole of $\lambda$ exactly with elementary computations such as (6).

Nevertheless, despite the apparent efficacy for tuning, there are of course some downsides.

- The handling of complex poles requires either complex number algebra or more involved real algebra [27], which could mitigate against easy coding and acceptance.
- Constraint handling was not tackled explicitly and again, could be somewhat more involved than discussed in Section 2.4.
- For higher order systems, there was an implicit need to select all the closed-loop poles, thus rendering this almost equivalent to a standard pole-placement approach. In general, it is not obvious how to select the less dominant poles, which means the approach loses some of its attraction.

### 3.4. Laguerre PFC

A core weakness of the conventional PFC approach that has been well recognised in the mainstream MPC literature [23,25,28,29] is the restriction, in the predictions, to a finite number of input changes in the immediate horizon. Indeed PFC assumes a constant future input, which is the very extreme case. Such a restriction is reasonable for systems with simple under-damped dynamics and where open-loop speeds of response are satisfactory, but this restriction is a significant problem where the open-loop has more complex dynamics and/or there is a desire for the closed-loop to have faster poles.

Recognising that the input prediction parameterisations for open-loop MPC formulations should have some dynamics was proposed in [30], where Laguerre polynomials were considered. Parameterising input sequences around Laguerre polynomials makes good engineering sense as they are built around a *target pole*, which the user can easily choose and thus, while investigated more widely since the approach was also recently considered for PFC [14].

The core proposal is to replace the parameterisation of future inputs as follows:

$$\Delta \underset{\rightarrow}{u}(z) = \sum_i L_i(z)\eta_i \quad \text{or} \quad \underset{\rightarrow}{u}(z) = u_{ss} + \sum_i L_i(z)\eta_i \tag{14}$$

where $L_i(z)$ are the Laguerre polynomials; for PFC, only the first polynomial is used:

$$L_1(z) = 1 + pz^{-1} + p^2 z^{-2} + \cdots \tag{15}$$

The pole to be selected is clearly seen here as $p$ and would logically match the choice of $\lambda$ in the PFC target.

Some reflections on this proposal are as follows.

- In general terms, there is a closer (compared to conventional PFC) correspondence between the closed-loop poles and the target pole of $\lambda$ when the input is parameterised as in (14). Thus, on balance, one might prefer this algorithm.
- It was noted that it was better to use the parameterisation $\underrightarrow{u}(z) = u_{ss} + L_1(z)\eta$ as otherwise, the transients speed of response was severely impeded by the shape of the first Laguerre polynomial.
- Despite being effective for some systems and an improvement on the conventional PFC approach, there was still poorer consistency between the closed-loop poles and the target pole $\lambda$ than was satisfactory. Moreover, this approach did not help, or indeed does not have sufficiently tailored shaping, with difficult dynamics such as open-loop unstable poles and oscillatory modes.

### 3.5. Closed-Loop or Pre-Stabilised PFC

The most recent suggestion for modifying PFC builds on some ideas with the case by case modifications for the original algorithm [6] but implements this in a more systematic and effective manner. The basic idea is to recognise that where a system has poor open-loop dynamics, a simple feedback loop, often based solely on simple proportional control, can improve those dynamics [11,16,17]. As the efficacy of PFC is strongly linked to the nominal dynamics having a shape somewhat close to a well-damped system, this initial pre-stabilisation can be hugely effective in enabling PFC. The ultimate design is now a cascade structure, but given the inner loop is a simple classical design, this additional complexity is a very minor addition and can be viewed as essential for handling challenging dynamics.

A minor consequence of using a cascade structure is that the d.o.f. to be utilised in solving (4) is now the input to the inner loop rather than the input signal (see Figure 2). In the early works, this variable has been parameterised as a single constant, but equally, we could make use of parameterisations such as in Section 3.4. Ironically, the use of a cascade loop has an insignificant impact on the constraint handling aspects as, albeit the parameters in the replacement for (8) are slightly different, the number of rows and treatment is similar.

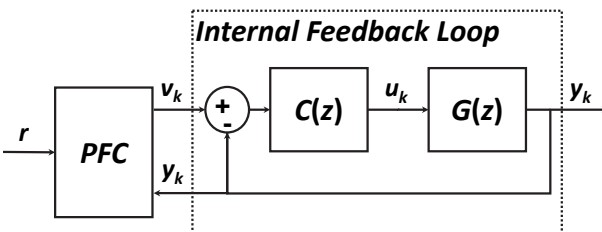

**Figure 2.** Cascade structure with PFC.

In summary, early reflections on using a simple cascade structure are:

- Final tuning outperforms conventional PFC for a range of dynamics and notably deals effectively with challenging dynamics such as under-damped processes and open-loop instability.
- Enables simple guarantees of recursive feasibility, that is, effective constraint handling; something other methods largely struggle with.
- The main downside is that, for systems with very challenging open-loop dynamics, it may not be straightforward to determine a simple classical control law such as PID for the inner loop. Of course, one could argue that such cases are irrelevant if we are considering PFC as a competitor to simple classical control. Moreover, conventional PFC had no answers to this either, and more expensive alternatives may be required.

### 3.6. Dealing with Lag in the Closed-Loop Responses

Notwithstanding the desire for the closed-loop pole to match the target pole, it was noted recently that there was a fundamental weakness in the definition of control law (4), which almost inevitably introduces some lag into the closed-loop responses [15]. This

paper will not discuss that issue in detail as, in effect, it relates primarily to the feedforward component of the control law and thus could be considered as a separate issue from controller tuning. Feedforward information needs to be integrated carefully or it could cause unexpected distortions in the behaviour.

In simple terms, some form of memory of the target and disturbance estimate information needs to be integrated into (4) and the implied feedforward to ensure consistency of planning from one sample to the next. Where the system response differs from a 1st order response in fast transients, as almost all real systems will, the nominal PFC law of (4) does not recognise this inconsistency adequately and thus may not behave as well as it could.

## 4. Comparison of Alternative PFC Approaches

This section presents several numerical examples with varying dynamics of the alternative approaches. These are used as evidence for some stronger conclusions given at the end of the section. Two aspects are considered to be at the core for a PFC algorithm to be successful:

1. The tuning needs to be intuitive, that is, not requiring expert involvement. Here, we take that to mean that the user can choose the desired pole $\lambda$ (in effect a closed-loop settling time) and the algorithm will deliver something close to that.
2. The coding requirements should be simple enough for interrogation and management by local staff, thus again cutting the reliance on expensive consultants. The code must not rely on expensive optimisers and use only the simplest coding constructs.
3. It is accepted that effective constraint handling is non-trivial even for PID approaches, and thus, the code for this part will inevitably be more than a few lines. However, it is similar for all the approaches, as mentioned in Section 2.4 and, thus, not presented.

Due to bullet point 1 above, we will consider three approaches: (i) conventional PFC (Section 2—denoted CPFC; (ii) Laguerre PFC (Section 3.4—denoted LPFC) and (iii) closed-loop PFC (Section 3.5—denoted CLPFC).

### 4.1. Case Studies

In line with earlier work [22,31], it is appropriate to use some industrially motivated case studies to benchmark the different approaches and numerical comparisons. These are selected to cover a range of challenging dynamics such as integrators, instability and oscillatory modes. These are summarised next.

**Boiler Level Control:** A typical model for this process [32] is usually a first-order system with an integrator, output water level and the input water flow rate; a representative model is:

$$G_1 = \frac{0.1z^{-1} + 0.4z^{-2}}{(1 - 0.8z^{-1})(1 - z^{-1})} \tag{16}$$

**Depth Control of Unmanned Free-Swimming Submersible (UFSS):** The depth of an unmanned submarine can be controlled by deflecting its elevator surface, whereby the vehicle will rotate about its pitch axis; the associated vertical forces due to the water flow enable the vehicle to sink or rise. Typical dynamics include one stable pole and two complex poles with stable zeros [33]. Thus, a representative model is given by $G_2$ below.

$$G_2 = \frac{0.85z^{-1} - 1.5z^{-2} + 0.85z^{-2}}{(1 - 0.6z^{-1})(1 - 1.6z^{-1} + 0.8z^{-2})} \tag{17}$$

with output pitch angle (rad) and input elevator deflection (m).

**Temperature Control of Fluidised Bed Reactor:** A fluidised bed reactor may involve exothermic reactions and thus open-loop instability. The reactor bed temperature (°C) is controlled with a coolant flow rate (m³ s⁻¹); thus, one obvious risk is that changes in

flow rate can easily trigger rapid divergence in the temperature. A representative model to capture the core dynamics for control design includes at least one stable pole and one unstable pole [34]; a typical example is given below as $G_3$.

$$G_3 = \frac{0.2z^{-1} - 0.26z^{-2}}{(1 - 0.9z^{-1})(1 - 1.5z^{-1})} \tag{18}$$

*4.2. Formation of Pre-Compensated Prediction Model for CLPFC*

For the integrator system $G_1$, a proportional controller $C(z) = 0.0461$ implemented as shown in Figure 2 results in the pre-compensated dynamics $T_1 = \dfrac{0.004615z^{-1} + 0.01846z^{-2}}{1 - 1.795z^{-1} + 0.8185z^{-2}}$ with stable poles at $0.8975 \pm \mathrm{j}0.1140$.

For $G_2$, a PI controller has to be tuned as P and PD controllers are incapable of stabilising the oscillatory open-loop behaviour. Therefore, $C(z) = 0.0038 + \dfrac{0.0423}{z - 1}$ is implemented, providing $T_2 = \dfrac{0.0032z^3 + 0.027z^{-1} - 0.055z^{-2} + 0.033z^{-3}}{1 - 3.20z^{-1} + 3.98z^{-2} - 2.30z^{-3} + 0.51z^{-4}}$ with stable poles at $0.822 \pm \mathrm{j}0.088$ and $0.78 \pm \mathrm{j}0.3840$.

For the unstable system $G_3$, the PID controller fails to stabilise; therefore, the inner loop has been tuned using a pole-placement method discussed in [35]. With $C(z) = \dfrac{22.914z^{-1} - 20.622z^{-2}}{1 - 4.55z^{-1} + 1.66z^{-2}}$, a pre-stabilised prediction model with stable poles at 0.9, 0.667 and 0.4 is obtained: $T_3 = \dfrac{4.58z^{-1} - 10.082z^{-2} + 5.36z^{-3}}{1 - 1.967z^{-1} + 1.227z^{-2} - 0.24z^{-3}}$.

In the following analysis, $T_1$, $T_2$ and $T_3$ designed above will be used with CLPFC, whereas both CPFC and LPFC will utilise the open-loop dynamics $G_1$, $G_2$ and $G_3$ for decision-making.

*4.3. Tuning Efficacy and Closed-Loop Performance Comparison*

Closed-loop performance of the three PFC approaches is evaluated with regards to the tuning efficacy for both short and long coincidence horizons. For each case study, the selected controller parameters $\lambda$ and $n$ are tabulated in Table 1. The simulation results are shown in Figures 3–5, and the key observations are:

- CLPFC clearly demonstrates a much stronger link between the actual and the desired closed-loop performance, even with longer coincidence horizons. This is important since long horizons may be necessary for better loop shaping against external perturbations, as pointed out in [17]. Furthermore, the control effort with CLPFC is smooth and the least aggressive as opposed to both CPFC and LPFC, which is crucial for constraint adherence in practice.
- The input parametrisation with CPFC and LPFC for small horizon apparently works with the relatively less difficult system $G_1$; nonetheless, both alternatives fail to handle the more challenging dynamics of $G_2$ (oscillations) and $G_3$ (non-minimum phase, instability). In contrast, the re-parametrisation due to the inner loop in CLPFC warrants a faster and smoother output convergence even in difficult applications.
- The dominant closed-loop poles given in Table 1 mirror the closed-loop performance presented graphically. Apart from $G_1$, the actual poles are far away from the desired $\lambda$ for CPFC, even the LPFC tuning is less effective for $G_2$ and $G_3$. However, the closed-loop poles remain stable and much closer to the target pole with CLPFC.

In short, the CLPFC algorithm overcomes the tuning and performance weaknesses of the CPFC for difficult dynamic problems. Although LPFC has also shown better performance as compared to the conventional algorithm for difficult but stable systems, it appears less effective in controlling non-minimum phase and/or unstable open-loop dynamics.

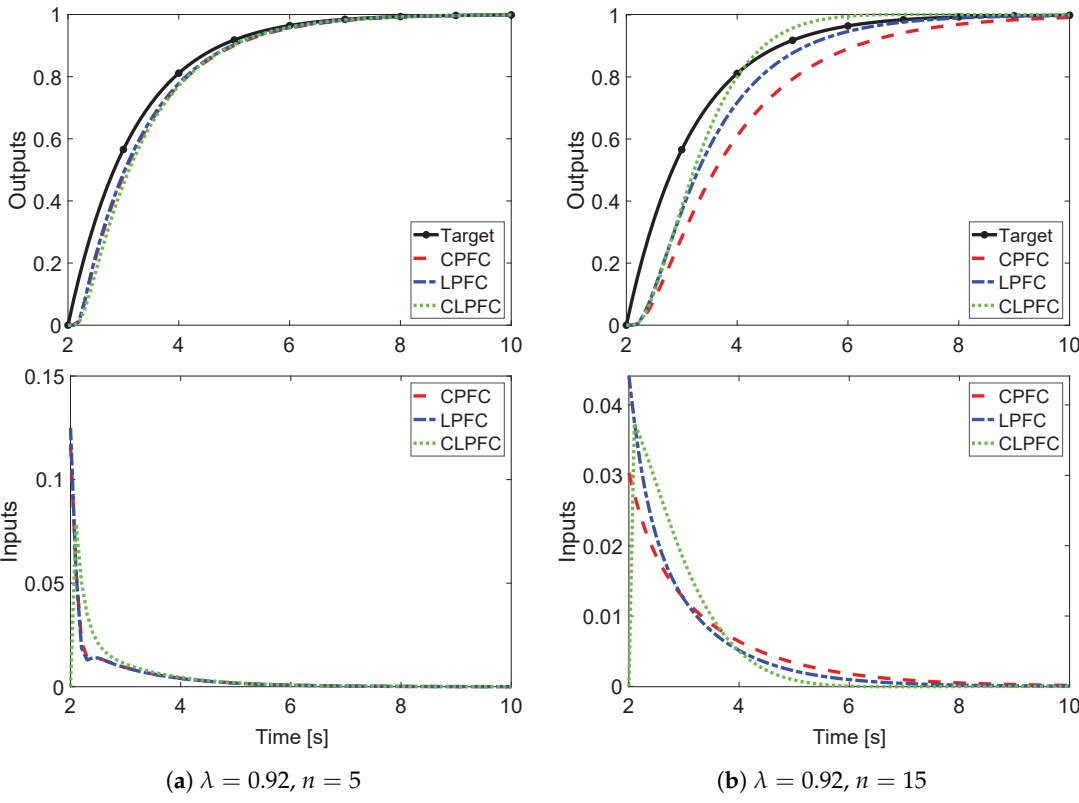

(**a**) $\lambda = 0.92$, $n = 5$

(**b**) $\lambda = 0.92$, $n = 15$

**Figure 3.** Comparison of closed-loop performance between CPFC, LPFC and CLPFC for $G_1$.

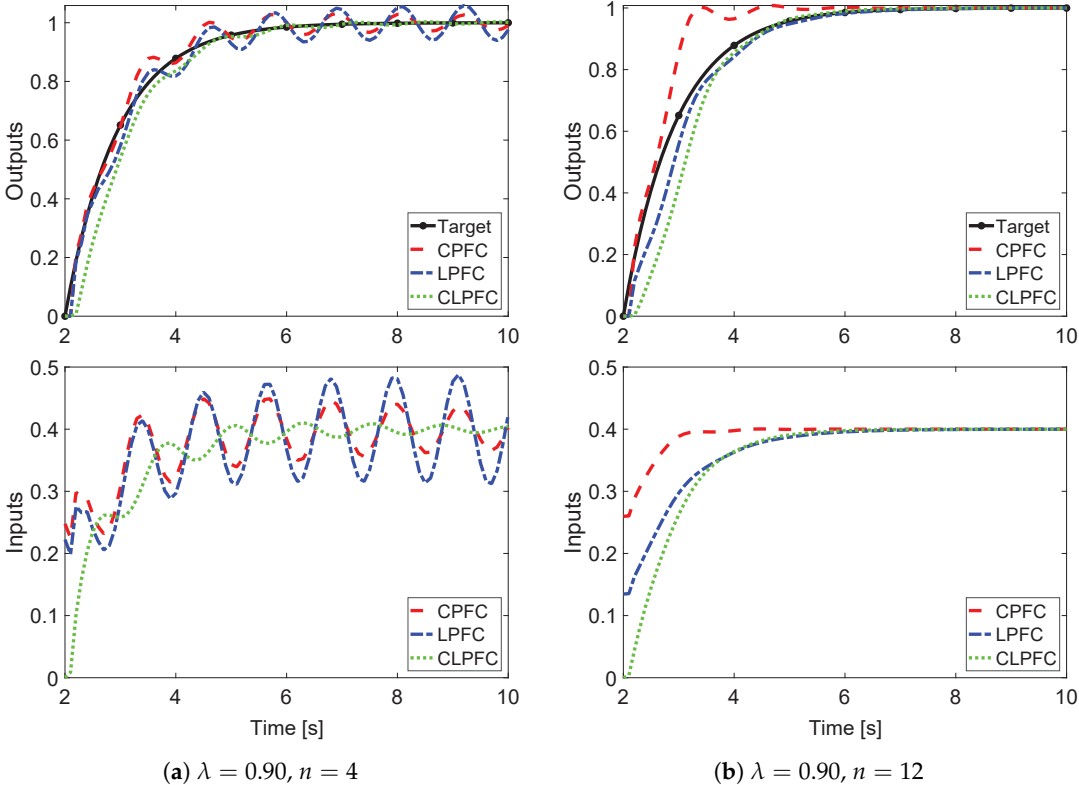

(**a**) $\lambda = 0.90$, $n = 4$

(**b**) $\lambda = 0.90$, $n = 12$

**Figure 4.** Comparison of closed-loop performance between CPFC, LPFC and CLPFC for $G_2$.

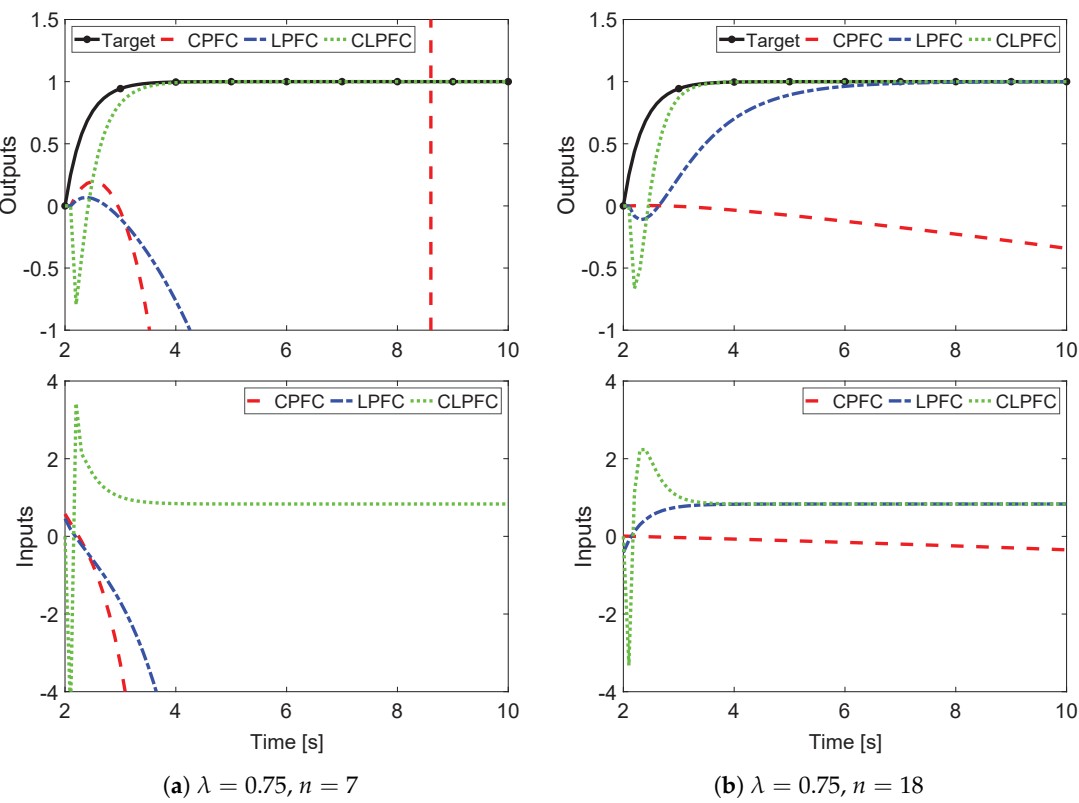

**(a)** $\lambda = 0.75$, $n = 7$        **(b)** $\lambda = 0.75$, $n = 18$

**Figure 5.** Comparison of closed-loop performance between CPFC, LPFC and CLPFC for $G_3$.

**Table 1.** Dominant closed-loop poles.

|  |  |  | **CPFC** | **LPFC** | **CLPFC** |
|---|---|---|---|---|---|
| $G_1$ | $\lambda = 0.92$ | $n = 5$ | $0.92$ | $0.92$ | $0.92, 0.53$ |
|  |  | $n = 15$ | $0.94, 0.75$ | $0.92, 0.72$ | $0.88 \pm \text{j}0.05$ |
| $G_2$ | $\lambda = 0.90$ | $n = 4$ | $0.88, 0.85 \pm \text{j}0.51$ | $0.9, 0.86 \pm \text{j}0.52$ | $0.9, 0.85 \pm \text{j}0.47$ |
|  |  | $n = 12$ | $0.78, 0.73 \pm \text{j}0.4$ | $0.9, 0.77 \pm \text{j}0.4$ | $0.88, 0.68, 0.78 \pm \text{j}0.39$ |
| $G_3$ | $\lambda = 0.75$ | $n = 7$ | $1.16, 0.84$ | $1.05, 0.75$ | $0.9, 0.74$ |
|  |  | $n = 18$ | $1.005, 0.9$ | $0.9, 0.75$ | $0.9, 0.68, 0.39$ |

## 5. Conclusions

This paper has presented a consolidated review of the recently proposed modified PFC algorithms, specifically focusing on the tuning issues pertaining to challenging dynamic applications. Numerous proposals have surfaced in the past two decades to improve control functionality, at least theoretically, although sometimes at the price of increased complexity resulting in diminished practical appeal. Nevertheless, this paper focuses on two recent approaches, namely Laguerre PFC and closed-loop (or pre-stabilised) PFC, that are well-explored in the mainstream MPC literature with proven efficacy in theory and practice. These modifications mainly work by introducing a different parametrisation of the degree-of-freedom, which is necessary to induce more flexibility to handle difficult dynamics in the control law; such flexibility does not exist in the conventional PFC algorithm, which consequently is difficult to tune effectively for many cases. The industrial case studies presented herein have demonstrated a superior performance and parameter tuning efficacy from the closed-loop PFC, which overcomes the inherent deficiency of the conventional algorithm against difficult dynamic problems to a greater extent while retaining the associated simplicity and intuitiveness. Although the Laguerre PFC alone may not perform as effectively in such applications, as a future consideration, nonetheless,

it is expected to yield further performance improvement when utilised in conjunction with the closed-loop PFC formulation.

**Author Contributions:** This paper is a collaborative work between both authors. J.A.R. provided initial proposals and accurate communication of the concepts. M.S.A. developed the code and analysed the concepts in the case studies. Both authors have read and agreed to the published version of the manuscript.

**Funding:** This research received no external funding.

**Institutional Review Board Statement:** Not applicable.

**Informed Consent Statement:** Not applicable.

**Data Availability Statement:** Not applicable.

**Acknowledgments:** The second author would like to acknowledge the University of Sheffield for his PhD scholarship, which is funding his studies.

**Conflicts of Interest:** The authors declare no conflict of interest.

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
