# Peer review of "A Comparison of Tuning Methods for Predictive Functional Control"

_processes, doi:10.3390/pr9071140_

Round 1
Reviewer 1 Report
The paper is well-written. Following points are aimed at improving the presentation.
- I feel the title is somewhat misleading. As noted in the Conclusion, the paper is more about comparison of recent PFC modifications.
- Pg. 3, line 66-68. control law has been used to refer to both (4) and (6). The reference to (4) is confusing.
- The paragraph starting on line 100 (pg.4) is not clear. Why would someone use inappropriate shaping in predictive control?
- Last line on pg. 5: 50 lines of code in which language?
- Pg. 7, line 197: you mean 'proportional control'? Also, as noted in bullet point 3, the concept of 'simple' classical design may be not clear.
- The formatting of figures and table in the pdf needs improvement.
Besides, though the author contribution section does not mention the writing part, apparently JAR did most of the writing and, in my opinion, should be the first author.
Author Response
Thank you for spotting the areas where corrections and clarification were needed. We detail the corrections in turn below.
Query: I feel the title is somewhat misleading. As noted in the Conclusion, the paper is more about comparison of recent PFC modifications.
Response: We have modified the title as suggested.
Query: Pg. 3, line 66-68. control law has been used to refer to both (4) and (6). The reference to (4) is confusing.
Response: We have added a clarifying comment about eqn. (6)
Query : The paragraph starting on line 100 (pg.4) is not clear. Why would someone use inappropriate shaping in predictive control?
Response: This is because in general the literature does not examine this issue carefully so many authors simply do not realise this is what they are doing. A clarifying comment has been added.
Query : Last line on pg. 5: 50 lines of code in which language?
Response: Have added the example of C and Python, but would apply to many similar languages.
Query: Pg. 7, line 197: you mean 'proportional control'? Also, as noted in bullet point 3, the concept of 'simple' classical design may be not clear.
Response: Added clarification as requested in both places.
Query: The formatting of figures and table in the pdf needs improvement.
Response: Apologies the table did have some silly typos we have corrected. We have ensured the figures largely fit into the column width, but assume the journal will format as required anyway in their publishing tool.
Query: Besides, though the author contribution section does not mention the writing part, apparently JAR did most of the writing and, in my opinion, should be the first author.
Response: OK, we have swapped the authors’ order.
Reviewer 2 Report
Τhis work summarizes and consolidates the work of the past decade which has focused on proposing more effective tuning approaches while retaining the core attributes of simplicity and low cost. The authors focus on the tuning issues pertaining to challenging dynamic applications. Specifically, the authors focus on predictive functional control (PFC) which is a fast and effective controller that is widely used. They concentrated on two recent approaches, namely Laguerre PFC and Closed-loop or Prestabilised PFC, that are well-explored in the mainstream MPC literature with proven efficacy in theory and practice. Nevertheless, the core advantages of simplicity and low cost come alongside weaknesses in tuning efficacy. The work finishes on the more effective approaches and links to context.
The paper addresses a topic posing theoretical challenges and having practical significance. It is methodologically correct. The paper is suitable for publication.
The literature review in introduction is thorough and it is very well written, however some additional references listing below regarding control algorithms for elimination the response of structure subjected to earthquake excitation could be added in the introduction.
- Nikos G. Pnevmatikos, Charis J. Gantes, “The influence of time delay and saturation capacity in control of structures under seismic excitations” Smart Structures and systems, An International Journal, Vol. 8, No. 5, pp 479-490, 2011.
- Nikos G. Pnevmatikos, “New strategy for controlling structures collapse against earthquakes”. Natural Science, Vol.4, pp. 667-676, 2012, doi:10.4236/ns.2012.428088.
Author Response
Thank you for the overview and comments. Our response to the main query is below.
Query: The literature review in introduction is thorough and it is very well written, however some additional references listing below regarding control algorithms for elimination the response of structure subjected to earthquake excitation could be added in the introduction.
Response: We have looked at the references as suggested and feel that the focus and control methods of those are somewhat different from that in our paper and thus their inclusion could be confusing to the reader, especially in the absence of a much longer introduction covering a far broader selection of the literature and control methodologies. We do not feel this would add value to the current work which very much focuses on PFC.